# Diagnosis, Investigation and Management of Patients with Acute and Chronic Myocardial Injury

**DOI:** 10.3390/jcm10112331

**Published:** 2021-05-26

**Authors:** Caelan Taggart, Ryan Wereski, Nicholas L. Mills, Andrew R. Chapman

**Affiliations:** 1BHF Centre for Cardiovascular Science, University of Edinburgh, Edinburgh EH16 4SA, UK; c.taggart@ed.ac.uk (C.T.); Ryan.Wereski@ed.ac.uk (R.W.); nick.mills@ed.ac.uk (N.L.M.); 2Usher Institute, University of Edinburgh, Edinburgh EH16 4UX, UK

**Keywords:** myocardial injury, myocardial infarction, universal definition

## Abstract

The application of high-sensitivity cardiac troponins in clinical practice has led to an increase in the recognition of elevated concentrations in patients without myocardial ischaemia. The Fourth Universal Definition of Myocardial Infarction encourages clinicians to classify such patients as having an acute or chronic myocardial injury based on the presence or absence of a rise or a fall in cardiac troponin concentrations. Both conditions may be caused by a variety of cardiac and non-cardiac conditions, and evidence suggests that clinical outcomes are worse than patients with myocardial infarction due to atherosclerotic plaque rupture, with as few as one-third of patients alive at 5 years. Major adverse cardiovascular events are comparable between populations, and up to three-fold higher than healthy individuals. Despite this, no evidence-based strategies exist to guide clinicians in the investigation of non-ischaemic myocardial injury. This review explores the aetiology of myocardial injury and proposes a simple framework to guide clinicians in early assessment to identify those who may benefit from further investigation and treatment for those with cardiovascular disease.

## 1. Introduction

The Fourth Universal Definition of Myocardial Infarction (UDMI) is an international expert consensus document which aims to promote a better understanding of the aetiology of myocardial injury and infarction, and to encourage a systematic approach to the subsequent investigation, treatment and clinical outcomes. This new definition challenges physicians to always consider the mechanisms of injury to ensure accurate diagnosis and treatment.

## 2. Classification of Myocardial Injury and Infarction

Myocardial injury is an umbrella term which describes an elevation in cardiac troponin concentration with at least one value above the 99th percentile upper reference limit (URL). This term can be applied irrespective of aetiology and can be classified into acute or chronic myocardial injury on the basis of a dynamic change in cardiac troponin concentration. Typically, a relative change in troponin concentration of 20% is used [1].

The diagnosis of myocardial infarction is applied when acute myocardial injury occurs in a patient with symptoms or signs of myocardial ischaemia on the electrocardiogram (ECG), or new imaging evidence of a regional wall motion abnormality in a coronary territory. Myocardial infarction is further stratified into five main subtypes (Figure 1) [1,2]. Type 1 myocardial infarction is due to atherosclerotic plaque rupture, leading to turbulent blood flow, platelet aggregation and coronary artery occlusion, with subsequent myocardial ischaemia and infarction. This phenotype is well understood, with evidence-based guidelines to inform both primary and secondary prevention which has been shown to improve clinical outcomes [3]. Type 2 myocardial infarction is due to oxygen supply and demand imbalance in the absence of atherosclerotic plaque rupture. Typically, this occurs in the context of a physiological stressor such as tachyarrhythmia, hypoxia, or hypotension. Clinical outcomes are significantly worse than type 1 myocardial infarction. This can be explained in part by the age and co-morbidities of the patients affected, but also may reflect suboptimal treatment of underlying coronary and structural heart disease [4,5,6]. Type 3 myocardial infarction occurs in the setting of sudden death prior to biomarker sampling. Type 4 myocardial infarction events occur due to percutaneous coronary intervention (4a) or as a result of stent thrombosis (4b) or in-stent restenosis (4c). Type 5 myocardial infarction occurs after cardiac surgery.

## 3. Detecting Myocardial Injury in Clinical Practice

The clinical application of cardiac biomarkers for the diagnosis of myocardial infarction has evolved from the use of non-specific indicators of muscle ischaemia and breakdown, including creatine kinase myocardial band (CK-MB), myoglobin, lactate dehydrogenase (LDH) and aspartate aminotransferase (AST), to the use of cardiac troponin as the only recommended biomarker [3,7]. Cardiac troponin exists as a complex in a series of proteins that form the thin filament in muscle. Troponin binds with calcium which allows bonding of actin and myosin, responsible for skeletal and cardiac muscle contraction [8]. The identification of cardiac-specific troponin I and T isoforms facilitated the development of biochemical assays which are highly sensitive and specific for cardiomyocyte injury. Both subtypes are released into the bloodstream in response to injury, but are not specific to an ischaemic aetiology [9,10].

Cardiac myocytes may release troponin into the blood plasma through a variety of mechanisms beyond cell necrosis and membrane rupture [11], but the pathophysiology is poorly understood, with most evidence existing in vitro. Non-ischaemic mechanisms may occur indolently and explain the silent release of troponin in patients with chronic myocardial injury. Reported mechanisms include normal cardiac cellular turnover, apoptosis [12] and free bound cytoplasmic troponin release through membranous blebs [13] (Figure 2). To test whether transient ischaemia affects troponin concentration in practice, a recent study measured blood high-sensitivity cardiac troponin (hs-cTn) concentration after balloon inflation within the left anterior descending coronary artery in a population of patients with otherwise angiographically normal coronary arteries. Troponin elevation was observed in all three high-sensitivity troponin assays just fifteen minutes after balloon occlusion for only 30 s, and this was elevated above the 99th percentile in 25% of patients sampled with hs-cTn T, fulfilling the biochemical criteria or myocardial infarction [14]. This demonstrates the remarkable sensitivity of these assays, which are able to detect myocardial injury in the context of transient myocardial ischaemia without infarction.

Cardiac troponin assays have evolved from contemporary sensitive to the hs-cTn immunoassays widely utilised today [7]. These assays can detect troponin isoforms I and T in the majority of healthy individuals, with some able to measure concentrations in over 90% [15,16]. A number of assays are commercially available, with a recent global survey of 1902 medical centres across 23 countries, highlighting that 96% used cardiac troponin as the primary diagnostic biomarker for myocardial infarction [17]. It is important to note that each assay has its own performance characteristics and reference ranges which are not transferrable. Clinicians must be aware of the assay in use at their own centre. A recent study evaluated 12 of the most commonly used assays in an ethnically diverse universal sample bank of healthy patients. The authors aimed to define sex-specific 99th percentiles using each assay, and found that only 8 assays were able to measure cardiac troponin in more than 50% of healthy men and women, thus fulfilling the criteria to be defined as a true high-sensitivity assay [18]. The magnitude and change in troponin concentration over time may serve as a useful tool for identifying the underlying mechanism of myocardial damage with several studies revealing peak troponin to be significantly higher in type 1 myocardial infarction than type 2 myocardial infarction and myocardial injury [19,20].

## 4. Epidemiology of Acute and Chronic Myocardial Injury

Troponin elevation is common in conditions other than acute coronary syndromes. A number of studies in both selected and unselected populations suggest that non-ischaemic myocardial injury is the most common cause of elevation in cardiac troponin concentration, [9,16,17,18,19], with more recent studies discriminating between acute and chronic myocardial injury [21,22,23,24] (Table 1). In an observational cohort study of 22,589 patients who had hs-cTnT measured in a single emergency department (ED) in Sweden [22,23], 65% of those with elevated cardiac troponin concentrations had myocardial injury (30% acute and 35% chronic), with the remaining 35% having a diagnosis of myocardial infarction. 

The High-Sensitivity Troponin in the Evaluation of Patients with Suspected Acute Coronary Syndrome (High-STEACS) trial evaluated 48,282 consecutive patients with suspected acute coronary syndrome tested using a hs-cTnI assay and demonstrated that of the 9115 patients with elevated cardiac troponin concentrations, 33% had non-ischaemic myocardial injury (18% acute and 12% chronic), with the remaining 67% having a diagnosis of myocardial infarction (50% type 1 and 17% type 2 myocardial infarction). In a further study of unselected consecutive patients who attended the emergency department (ED) and underwent blood testing on clinical indication, hs-cTnI measurements were conducted in all patients as a suppressed test. This found elevated cardiac troponin concentrations in 13.7% (114/1054), of which 96% were due to non-ischaemic myocardial injury [21,31].

In an observational cohort study of 1640 consecutive unselected patients in which troponin sampling was performed on clinical indication in the ED, 56% of the 497 patients with elevated troponin concentrations had non-ischaemic myocardial injury [25]. The wide variation in incidence of myocardial injury could be explained by the difference in patient selection for testing, study inclusion criteria and diagnostic adjudication with a higher incidence of non-ischaemic myocardial injury seen in unselected patient cohorts.

## 5. Mechanisms of Myocardial Injury

### 5.1. Acute Myocardial Injury: Cardiac Mechanisms

It is now widely recognised that a plethora of cardiac and non-cardiac conditions may be responsible for both acute and chronic myocardial injury (Figure 3). Cardiac causes are common with tachyarrhythmia the most frequently observed cause of acute myocardial injury in clinical practice. Another common cause is acute decompensated heart failure, where elevated troponin levels associated with worse outcomes [32]. Myocardial injury is frequently observed following the direct insult of cardiac surgery [33], cardiac trauma or contusions as a consequence of cardiopulmonary resuscitation [34]. Myopericarditis [35,36] and less commonly endocarditis are associated with acute myocardial injury with the mechanism for the latter thought to be due to associated left ventricular dysfunction. This was suggested in a study of 97 patients with endocarditis and biomarker sampling at 1 and 7 days. Concentrations were higher in those with heart failure and valvular dysfunction. This study also revealed BNP as well as troponin concentration to be good predictors of mortality [37].

Takotsubo or stress cardiomyopathy is a poorly understood condition thought to be caused by catecholamine induced coronary vasospasm. Despite this presumed ischaemic aetiology, it is not considered a cause of myocardial infarction. Characteristically, there is ballooning of the apical segment of the heart. These patients, however, quite often only display a modest rise in troponin level in comparison to the degree of left ventricular dysfunction and studies have suggested patients with high troponin levels at presentation are unlikely to have Takotsubo cardiomyopathy [38,39]. Comparatively in myocarditis, troponin elevation may be more marked and persistently elevated [40]. Although troponin elevations have been demonstrated to have good discrimination for the diagnosis of myocarditis and are recommended by expert consensus, the best diagnostic tool is cardiac magnetic resonance imaging (cMRI) for those without complications such as cardiogenic shock or acute heart failure, or endomyocardial biopsy for those patients with decompensation where cMRI is not feasible or there is a high degree of diagnostic uncertainty [40]. Acute aortic syndromes including dissection can cause myocardial injury due left ventricular overload, coronary dissection, or in some cases a dissection flap causing coronary artery ostial disruption which may present with ST-segment elevation [41].

### 5.2. Acute Myocardial Injury: Non-Cardiac Mechanisms

Non-cardiac causes of acute myocardial injury include pulmonary embolism (PE), with myocardial injury occurring due to hypoxia [35,42], or haemodynamic effects resulting in right heart strain. The European Society of Cardiology (ESC) guidelines identify troponin as a prognostic marker in patients with PE to help identify those at high risk of death and consideration of thrombolysis therapy [43,44]. There are neurological causes of myocardial injury including stroke [45] and sub-arachnoid haemorrhage [46]. The exact mechanism of troponin elevation has not been characterised. However, the accompanied hypertension, catecholamine release and vasoconstriction in these conditions are likely to be contributary with dynamic ECG changes often seen. Hypertension has been further demonstrated to be linked to myocardial injury in a secondary analysis of 5251 patients in the Atherosclerosis Risk in Communities Study (ARIC), with isolated systolic blood pressure positively correlating with cardiac troponin concentration [47]. The secondary effects of sepsis have been shown to lead to myocardial injury [48,49], with the profound haemodynamic consequences of severe infection implicated in addition to direct myocardial injury [35].

Other systemic stressors such as non-cardiac surgery [50] have been associated with acute myocardial injury and have been associated with post procedural mortality. Chemotherapeutic agents such as anthracyclines and Herceptin can cause left ventricular dysfunction and associated myocardial injury. Cardiac biomarkers are often monitored in conjunction with echocardiography to facilitate early identification prior to established cardiac toxicity [51].

Myocardial injury has also been documented as a transient phenomenon after high-intensity exercise [52,53]. Troponin concentrations are higher after short intense periods of exercise than following prolonged exercise at lower intensity, and in healthy persons levels return to pre-exercise levels within 24 h. Elevation in both troponin I and T has been documented in athletes and non-athletes after major endurance events, with up to 43% of non-athletes having concentrations above the 99th percentile [54,55,56]. Historically, rhabdomyolysis, which can occur after intense exercise or muscle injury, has been positively correlated with elevated cardiac troponins [57]. However, a recent adjusted analysis revealed high levels of skeletal muscle breakdown not to be a relevant non-cardiac cause of acute myocardial injury [58]. Aengevaeren and colleagues evaluated 12 moderately to highly trained athletes shortly after participating in a marathon, of whom 7 (64%) had elevation in cardiac troponin I above the 99th percentile. They utilised advanced multi-modal cMRI and observed increased mean diffusivity and fractional anisotropy of myocardial tissue water from baseline which was correlated to troponin concentration. An increase in mean extracellular volume in cardiomyocytes of all athletes was also demonstrated. This study supports the theory of reversible myocardial injury through increased cardiomyocyte permeability [59]. A further study measured high-sensitivity troponin I in 725 people after 30–55 km of walking. They observed an increase in prevalence of participants with troponin concentrations over the 99th percentile from 1 *to* 9% after walking. During an average of 43 [23,24,25,26,27,28,29,30,31,32,33,34,35,36,37,38,39,40,41,42,43,44,45,46,47,48,49,50,51,52,53,54,55,56,57,58,59,60,61,62,63,64,65,66,67,68,69,70,71,72,73,74,75,76,77] months of follow up, patients with acute myocardial injury had a significantly increased risk of all-cause mortality or major adverse cardiovascular events (MACE) after adjustment for age, sex and cardiovascular risk factors (adjusted hazard ratio (aHR) 2.48; 95% Confidence Interval (CI) 1.29 to 4.78). Whilst this association may be due to unmeasured confounding or the presence of unrecognised cardiovascular disease, they suggest that troponin release during exercise may not always be a benign response [60].

### 5.3. Chronic Myocardial Injury: Cardiac Mechanisms

Chronic myocardial injury tends to reflect a steady disease state, and is frequently observed in chronic heart failure [61]. A meta-analysis of patients with elevated baseline cardiac troponin in chronic stable heart failure demonstrated an association with all-cause mortality (hazard ratio (HR) 2.85; 95% CI 2.02 to 4.03) and a higher rate of adverse cardiovascular outcomes (HR 2.38; 95% CI 1.63 to 3.49) when compared to similar populations in which there was little evidence of myocardial injury [62]. Other cardiac causes of chronic injury include cardiomyopathies [63,64] including infiltrative cardiomyopathies such as sarcoidosis, [65] haemochromatosis, amyloidosis [66] and hypertensive heart disease [67].

Valvular heart disease is also commonly associated with elevation in cardiac troponin concentration. The Early Valve Replacement Guided by Biomarkers of LV Decompensation in Asymptomatic Patients with Severe AS (EVoLVeD) (NCT:03094143) is a randomised controlled trial in patients with asymptomatic severe aortic stenosis aiming to optimise the timing of valvular surgery. In this trial, cardiac troponin is used as a screening marker of early cardiac fibrosis to identify patients for cardiac magnetic resonance imaging. Patients with evidence of mid-wall fibrosis are randomised to early valve replacement or watchful waiting.

The mechanisms responsible for chronic myocardial injury in patients with severe or diffuse coronary artery disease are poorly understood. The Dual Antiplatelet Therapy to Reduce Myocardial Injury (DIAMOND); (NCT:02110303) trial identified a cohort of patients with high-risk coronary plaque using coronary 18F-fluoride positron emission tomography/coronary computed tomography. In the subset of patients with elevated hs-cTn I concentrations ≥5 ng the addition of ticagrelor compared to placebo to standard preventative therapy did not reduce cardiac troponin levels over one year. This implies that subclinical plaque instability leading to ongoing micro-thrombosis is not the primary contributing cause of measurable troponin concentrations in these patients [68].

The West of Scotland Coronary Prevention Study (WOSCOPS) trial randomised a population of 6595 men with moderate hypercholesterolaemia to a placebo or pravastatin 40 mg. In a secondary analysis of this trial, troponin concentrations were predictive of coronary events and were modified by statin therapy, with the change in LDL at 1 year associated with future coronary risk independent of cholesterol lowering. Furthermore, those with decreasing troponin concentration by more than one-quarter, rather than increasing by more than one-quarter, had a five-fold lower rate of future coronary events in both the placebo group (HR 0.29; 95% CI 0.12 to 0.72 versus HR 1.95; 95% CI 1.09 to 3.49; *p* < 0.001 for trend) and the pravastatin group (HR 0.23; 95% CI 0.10 to 0.53 versus HR 1.08; 95% CI 0.53 to 2.21; *p* < 0.001 for trend). This suggests there may be value of repeated sampling in patients with chronic troponin elevation to target those at high risk of future events and monitor response to treatment [69].

### 5.4. Chronic Myocardial Injury: Non-Cardiac Mechanisms

Non-cardiac aetiologies of chronic myocardial injury include pulmonary hypertension, toxins and diabetes mellitus [4,70]. However, probably the most frequently observed non-cardiac cause of chronically elevated cardiac troponin levels is in patients with chronic renal disease. Cardiovascular disease is common in patients with chronic kidney disease, with half of all deaths in end-stage renal failure due to cardiovascular events [71]. However, the aetiology of troponin elevation is unclear and may reflect either decreased renal clearance or increased myocardial release of troponin.

The mechanisms by which troponin is broken down and excreted in humans remains to be fully understood and despite its importance, little research has been conducted in vivo. As with most large proteins, it is hypothesised that the predominant process is by scavenger receptor-mediated clearance [72]. This extra-renal clearance has been demonstrated in vitro during high levels of circulating cardiac troponin using experimental models of rats with and without renal function. Cardiac troponin T concentration still reduced over time after intravenous infusion of cardiac extracts in rats without kidney function. Increased clearance of cardiac troponin T was only seen when levels were low [73]. This can likely be explained by scavenger receptors having reduced affinity for troponin at such low concentrations and smaller degradation products of troponin readily passing through the glomerular membrane. A further study using radiolabelled cardiac troponin in rats revealed uptake in both the kidneys and liver, with liver uptake inhibited through inhibition of endocytosis, suggesting both renal and extra-renal systems of clearance [74]. These studies may explain the correlation between those with reduced renal function and slightly elevated concentrations of circulating cardiac troponin. However, animal models may not be reflective of human physiology.

The Chronic Renal Insufficiency Cohort Study (CRIC) enrolled 3664 patients with chronic renal disease and performed cardiac biomarker testing in all. High-sensitivity cardiac troponin T concentrations were found to be independently associated with all-cause mortality (HR 1.62; 95% CI 1.48 to 1.78). A further analysis of this cohort modelled to adjust for age, sex, presence of cardiovascular disease and cardiac risk factors revealed increased risk of cardiovascular death (aHR 1.87; 95% CI 1.65 to 2.11). There may be merit in using cardiac troponin concentration to stratify cardiovascular risk in those with chronic renal failure, identifying those with high circulating cardiac troponin concentrations for aggressive primary and secondary preventative therapies for cardiovascular disease.

## 6. Clinical Outcomes in Patients with Myocardial Injury

### 6.1. Mortality Outcomes

Outcomes in patients with acute and chronic myocardial injury are poor, with up to 72% of patients dead at five years [5] (Figure 4). In nearly all studies that have classified patients according to the Fourth Universal Definition of Myocardial Infarction, death from any cause was higher in patients with myocardial injury compared to type one myocardial infarction (Table 1). Outcomes are likely to be strongly influenced by the prognosis of the underlying condition responsible for myocardial injury and these patients tend to be older, with more cardiac and non-cardiac co-morbidities [19,21,22,23,24,70,71].

Sarkisian and colleagues evaluated a consecutive patient cohort in Denmark who had troponin measured on clinical indication. They found that 29% (1089/3762) of all patients had non-ischaemic myocardial injury. In this cohort, 59% (645/1089) of patients with myocardial injury were dead at 3.2 years, which was significantly higher than those with type 1 myocardial infarction (29%) but not type 2 myocardial infarction (63%). Patients with myocardial injury were more likely to present with dyspnoea and were older, and there was a higher incidence of heart failure with lower peak troponin concentrations compared to those with type 1 myocardial infarction.

Smilowitz et al. studied a similar cohort of patients with troponin concentration measured on clinical indication. They found non-ischemic myocardial injury in 175/710 (25%) of patients with a mean age of 74.9. However, in this cohort no difference in all cause death was observed between those with type 1, type 2 myocardial infarction and myocardial injury (30%, 31%, and 30%, respectively) at a median of 1.8 years. Differences in outcomes may perhaps be explained by selection, as this cohort was nearly all male, with a much higher proportion of type two myocardial infarction and less myocardial injury. This cohort had a higher incidence of previous heart failure in the type 1 group and no difference in incidence of chronic kidney disease, diabetes and chronic obstructive pulmonary disease between all groups. When evaluating outcome studies in patients with myocardial injury attention to the baseline characteristics, underlying concomitant disease profiles, treatment and diagnostic adjudication must be considered as these variables are likely to differ between both research and clinical sites.

### 6.2. Cardiovascular Outcomes

Most studies demonstrate that cardiovascular death and recurrent myocardial infarction are more common in patients with type 1 myocardial infarction, but patients with both acute and chronic myocardial injury have a substantially higher risk than those without myocardial injury [21,22,23,75].

Sandoval et al. evaluated 1640 unselected patients presenting to the emergency department using a contemporary sensitive cardiac troponin assay. They report a 19% MACE rate at 180 days in the 280 patients with myocardial injury. Interestingly, the only independent predictors for death in this group were congestive heart failure (HR 2.04, 95% CI 1.19 to 3.51) and peak cardiac troponin (HR 1.74, 95% CI 1.18 to 2.55) [25]. In the High-STEACS study, the risk of subsequent cardiovascular death or myocardial infarction in patients with acute myocardial injury and chronic myocardial injury was four-fold higher than in those without myocardial injury (cause specific hazard ratios 4.38, 95% CI 3.80 to 5.05) and (3.88, 95% CI 3.31 to 4.55), respectively.

Given the increased incidence of future MACE in patients with acute and chronic myocardial injury, further cardiac investigation may plausibly improve care in this population. There may be an opportunity for treatment optimisation whilst in hospital or at the point of discharge with the potential to modify outcomes in this group.

## 7. Investigation and Management Strategies for Acute Myocardial Injury

In the absence of formal guidelines, we propose a simple framework to guide clinicians when interpreting troponin results in patients who do not have myocardial infarction (Figure 5).

The primary goal of the assessing clinician should be to determine the correct diagnosis at the first consultation to guide subsequent investigation and treatment. After myocardial injury has been diagnosed on serial cardiac troponin testing, it is important to revisit the history and ensure there are no features consistent with myocardial ischaemia which could indicate myocardial infarction. In practice, differentiating acute non-ischaemic myocardial injury from type 2 myocardial infarction can be challenging. This often occurs as the classification of myocardial ischaemia is subjective on the basis of the clinical history and electrocardiogram. Difficulties in discriminating between acute myocardial injury and type 2 myocardial infarction can be exacerbated by atypical presentation of symptoms and non-specific ECG changes. Interestingly, patients with type 2 myocardial infarction and objective myocardial ischaemia on the electrocardiogram have been shown to have worse outcomes than those with only subjective ischaemic features [76]. Identification of regional ischaemia is important as it implies there may be a coronary mechanism, which may have important consequences for management.

A 12-lead electrocardiogram should be performed in all patients with myocardial injury. This is not only useful for exclusion of ischaemia, but may also indicate abnormalities suggestive of cardiomyopathy, brady/tachyarrhythmias or pulmonary hypertension and embolism which should be further investigated with monitoring or imaging. Haematology and biochemistry investigations should be considered including a full blood count, clotting screen, liver function tests, inflammatory markers, estimated glomerular filtration rate and electrolytes, with additional tests such as thyroid function and d-dimer considered if appropriate.

Further investigations should be guided by the history. In the acute setting, pleuritic pain may represent a multitude of pathologies and a plain chest radiograph may indicate infection or a primary lung pathology. Other causes such as myopericarditis should be considered by interrogating the electrocardiogram for ST segment or T-wave changes across different coronary territories. When pain occurs concomitantly with tachycardia and hypoxia then PE should be excluded, with risk stratification using the Geneva score, d-dimer and a CT pulmonary angiogram considered. Central chest pain which radiates through to the back may represent aortic dissection which is a surgical emergency and ECG-gated CT imaging should be performed if this condition is suspected. Other cardiovascular symptoms such as breathlessness, leg swelling, orthopnoea and reduced exercise tolerance may indicate heart failure which may have a valvular, structural, or chronic ischaemic cause. In this instance echocardiography should be considered. Palpitations and syncope may represent arrythmias in which case rhythm monitoring should be commenced. Detailed clinical examination is important as this may indicate signs of systematic illness or infection. An accurate medication history for both prescribed and not prescribed drugs is also essential as these may be causal for myocardial injury including previous chemotherapy agents and use of illicit substances such as methamphetamine or cocaine.

When there is a <20% change in troponin concentration on serial sampling then myocardial injury should be classified as chronic. Similar consideration of the underlying mechanism of myocardial injury is important, particularly the exclusion of structural or hypertensive heart disease, and treatment should be optimised such as the case in heart failure or chronic kidney disease.

In instances where cardiac imaging does not identify a clear cause of myocardial injury in otherwise healthy individuals, clinicians should always consider laboratory error and the possibility of analytical false positive results. This may occur due to heterophilic antibodies or interferents, which can occur in ~1 in 1000 patients. In addition to laboratory testing, clinicians should consider measuring an alternative commercially available troponin assay to determine if results are concordant. This is particularly relevant for future admissions to avoid unnecessary investigation or treatment.

Patients with acute and chronic myocardial injury are heterogenous, and an individualised management strategy is needed for each patient. Careful attention to the underlying mechanism of injury is important as a given treatment pathway may be advantageous for one patient but cause harm to another. For example, consider an elderly patient with a background of chronic kidney disease and known moderate aortic stenosis. They present with breathlessness and have stable troponin concentrations above the 99th percentile. Their last echocardiogram was within one month which demonstrated unchanged left ventricular function and similar aortic valve velocities to six months prior. Further investigation is unlikely to influence management and will increase the healthcare burden with no clear benefit to the patient, although it would be important to review the echocardiogram to ensure the assessment of severity is accurate as if severe, elevated cardiac troponin concentrations are independently associated with mid-wall fibrosis and may merit assessment for early valve replacement as outcomes in this group are much worse [77]. On the other hand, consider a 42-year-old patient who presents with symptoms of decreased exercise tolerance and shortness of breath. Identification of stable cardiac troponin concentrations above the 99th percentile is highly abnormal, and likely to be indicative of an undiagnosed cardiac pathology. Further investigations including a chest radiograph, electrocardiogram and transthoracic echocardiography are indicated to identify structural abnormalities and assess left ventricular function.

## 8. Conclusions

The implementation of highly sensitive troponin testing in an aging population has increased the prevalence of acute and chronic myocardial injury in practice. Myocardial injury is associated with worse outcomes irrespective of the underlying mechanism. Defining the underlying cause of myocardial injury is important and may have therapeutic implications that could change care and prognosis.

## Figures and Tables

**Figure 1 jcm-10-02331-f001:**
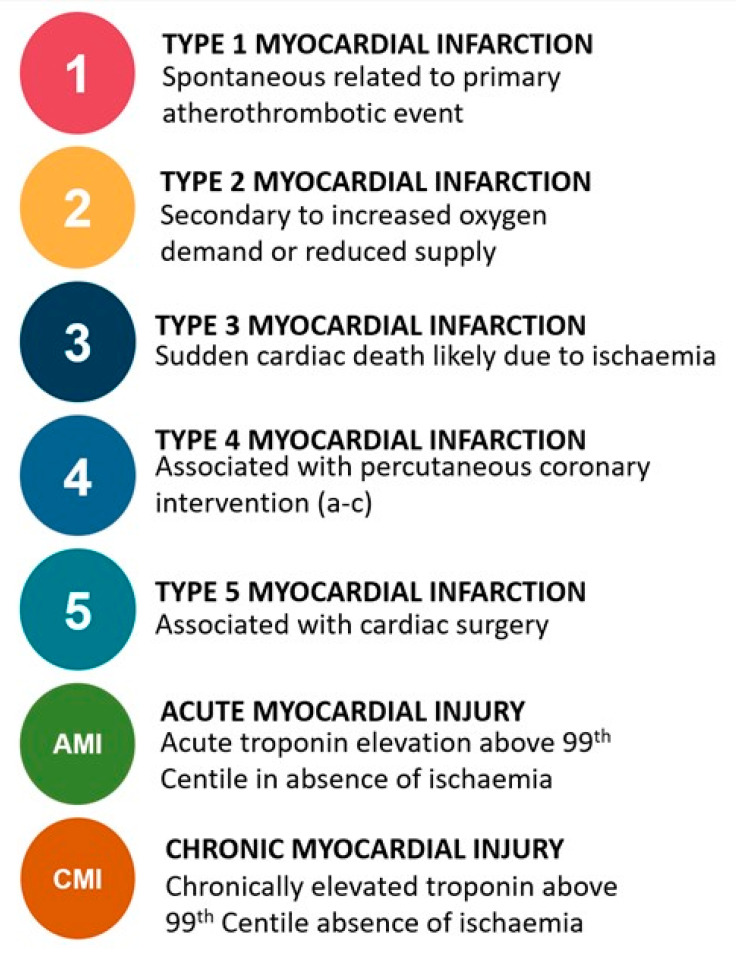
Sub-types of myocardial injury and infarction as per the Fourth Universal Definition of Myocardial Infarction.

**Figure 2 jcm-10-02331-f002:**
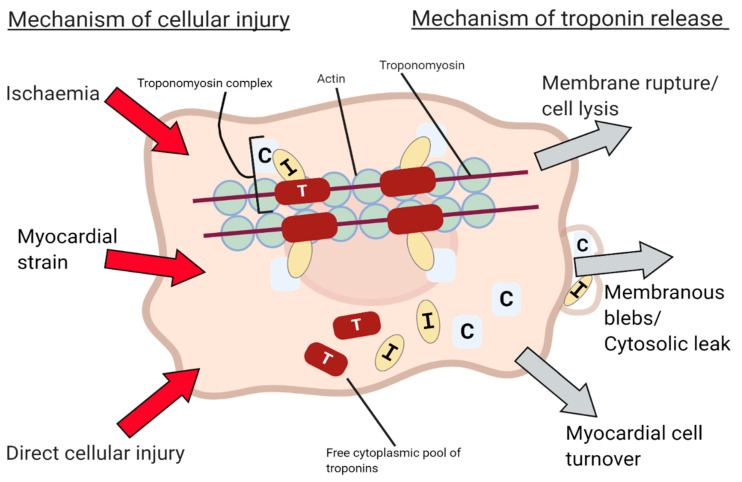
Proposed mechanisms of cardiac cell damage and troponin release.

**Figure 3 jcm-10-02331-f003:**
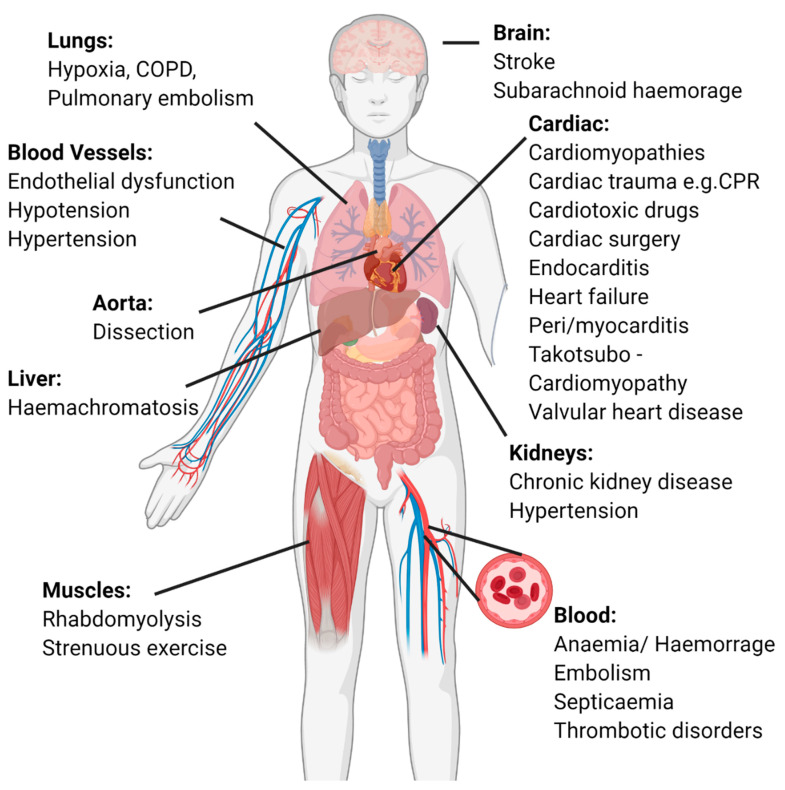
Systems-based approach to identifying causes of myocardial injury, listing common aetiologies in each system.

**Figure 4 jcm-10-02331-f004:**
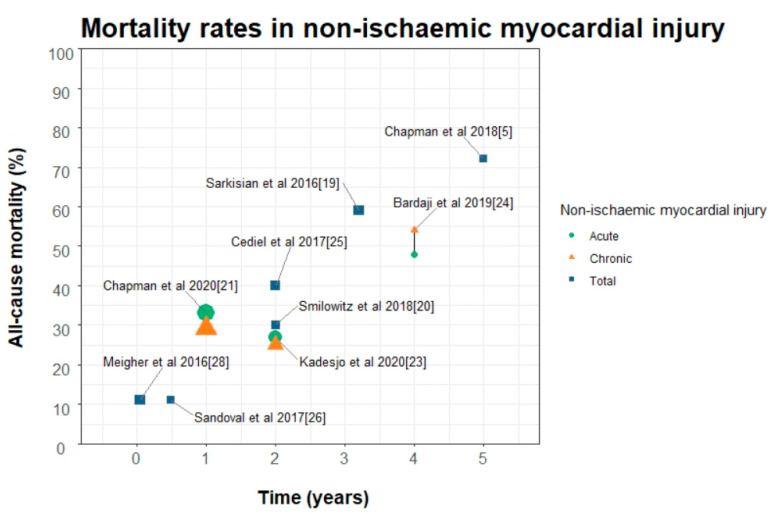
Mortality rates of non-ischaemic myocardial injury stratified into acute and chronic in studies from 2016 to 2020; titles are main study author and year as described in Table 1. Studies conducted before the 4th universal definition are displayed as total non-ischaemic myocardial injury, shape size is proportional to the number of study participants.

**Figure 5 jcm-10-02331-f005:**
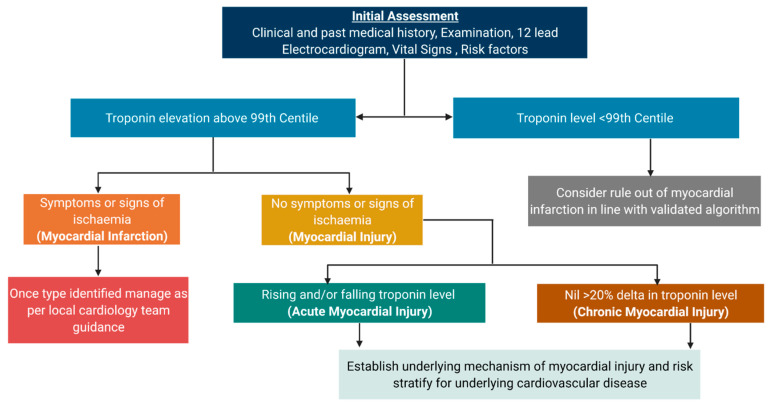
Taxonomy of myocardial injury and subclasses paired with basic investigation and management strategy.

**Table 1 jcm-10-02331-t001:** Table illustrating incidence of myocardial injury as per the universal definition over population subsets, proportion of non-ischaemic myocardial injury, cardiovascular and total mortality outcomes.

Study Lead Author and Year	Population Size	Total Incidence Myocardial Injury (Available for Adjudication)	Total Incidence Non-Ischaemic Myocardial Injury (% of Total Myocardial Injury)	Incidence Acute Myocardial Injury (%)	Incidence Chronic Myocardial Injury (%)	Patient Population	Total All-Cause Mortality Acute Myocardial Injury	Cardiovascular Outcomes and Mortality	Type 1 Myocardial Infarction Mortality (%)
4th Definition									
Chapman 2020 [21]	48,282	9115	2963 (33%)	1676 (18%)	1287 (14%)	Patients presenting to ED with chest pain.	1 year Acute: 33% Chronic: 29%	1 year CV death + MI Acute: 16% Chronic: 16%	1 year 14%
Kadesjö 2019 [22] Kadesjö 2020 [23]	22,589	3853	2491 (65%)	1144 (30%)	1347 (35%)	Patients presenting to ED with clinical indication, single centre with troponin >99th percentilecentile URL.	2 year Acute: 27% Chronic: 25% 4 year Acute: 53% Chronic: 52%	Median 4.0 ± 1.3 years CV death Acute: 19% Chronic: 20%	2 year 12% 4 year 35%
Bardají 2019 [24]	3701	-	368	261	107	Patients presenting to ED, clinical discretion to rule out ACS.	4 year Acute: 48% Chronic: 54%	4 year MACE Acute: 53% Chronic: 64%	N/A
3rd Definition									
Cediel 2017 [25]	3790	1010	440 (44%)	-	-	Retrospective cohort of patients presenting to ED with suspected ACS.	2 year 40%	-	2 year 19.7%
Sandoval 2017 [26]	1640	497	280 (56%)	-	-	Unselected patients presenting to the ED with troponin measured on clinical indication.	180 day 11% 2 year 26%	180 day MACE: 19%	180 day 8% 2 year 16%
Shah 2015 Chapman 2018 [5,27]	2122	2122	522 (25%)	-	-	Hospitalised patients with troponins taken for clinical indication.	1 year 37% 5 year 72%	1 year MACE: 18% 5 year 31%	1 year 16% 5 year 37%
Smilowitz 2018 [20]	Unknown	768	420 (55%)	-	-	Single centre with all troponin levels taken on clinical indication that were elevated.	In hospital 9% 2 year 30%	In hospital + 2 year follow up CV death 11%	In hospital 13% 2 year 30%
Meigher 2016 [28]	13,502	1283	458 (35.7%)	-	-	Single centre with patients presenting with suspected ACS.	Index hospitalisation 11%	-	Index hospitalisation 7%
Lee 2018 [29]	918	114	109 (96%)	-	-	Patients presenting to ED without suspicion of ACS.	Top quartile of troponin concentrations 1 year 37.7%	-	-
2nd Definition									
Sarkisian 2016 Lambrecht 2018 [19,30]	3762	1577	1089 (69%)	-	-	Hospitalised patients who had troponins taken as per clinical indication.	3.2 year median 59%	-	3.2 year median 39%

Abbreviations: ED, emergency department. ACS, acute coronary syndromes. MACE, major adverse cardiovascular events. CV death, cardiovascular death. MI, myocardial infarction.

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
