# Peer review of "Diagnosis, Investigation and Management of Patients with Acute and Chronic Myocardial Injury"

_jcm, 2021, doi:10.3390/jcm10112331_

Round 1

Reviewer 1 Report

Congratulations on this paper.

There is a minor mistake in one of the references. Should be:

Bardají A.; Bonet G.; Carrasquer A.; Gonz M.; Ali S.; Boqu C. Clinical Features and Prognosis of Patients with Acute and 480
Chronic Myocardial Injury Admitted to the Emergency Department. Am J Med. 2019, 132(5), 614–21

Reviewer 2 Report

What follows is my review of the manuscript entitled “Diagnosis, investigation and management of patients with acute and chronic myocardial injury”, by C. Taggart et al.  This is a review about patients with elevated troponin levels in the absence of myocardial ischemia.

First, it is worth mentioning that using the term “injury” could lead to confusion in some old practitioners, given that, long ago, the term injury was used to define EKG changes (injury current or pattern)  beyond the first few minutes of acute myocardial ischemia on patients that were progressing to acute myocardial infarction.

The authors provide a nice introduction, including the most current classification of myocardial infarction and the ways in which troponin could be present above normal limits without having myocardial ischemia.  The authors listed several blood tests that should be part of the initial screening of the patients with suspected myocardial injury.  Renal function panel should be included along them.

In the review, there is not even the slightest mention of possible false positive troponin result as an explanation for the laboratory abnormality.  I believe this should be mentioned as a possibility.  The discussion of management strategies is vague, in great part due to the fact that management still depends on the problem leading to troponin elevation, ie, tachyarrhythmia, aortic stenosis, pulmonary embolism, etc, etc.

Figures are very nice, simple, and provide a lot of information.

Reviewer 3 Report

In this review authors have tried to focus on diagnosis, investigation and management of patients with acute and chronic myocardial injury. However the article is interesting it brings no new knowledge to the field. 

Reviewer 4 Report

This is a review on acute and chronic myocardial injury, summarizing the central epidemiological, pathophysiological, and diagnostic aspects of these common conditions. The topic is of clear clinical relevance. The authors provide an interesting overview of the main diagnostic elements and propose an algorithm to guide management strategies in patients with elevated troponin levels. The paper is well written and accompanied by informative tables and figures. I have no specific issues to raise.

Round 2

Reviewer 3 Report

The article is interesting, however I still do not see anything new that it will bring to the field. Even Review article should carry some elements of new knowledge.